



# Historical and Future Weather Data for Dynamic Building Simulations in Belgium using the MAR model: Typical & Extreme Meteorological Year and Heatwaves

Sebastien Doutreloup[1], Xavier Fettweis[1], Ramin Rahif[2], Essam Elnagar[3], Mohsen S. Pourkiaei[4], Deepak
Amaripadath[2], Shady Attia[2]

[1] Laboratory of Climatology and Topoclimatology, Department de Geography, UR SPHERES, University of Liège, Belgium
[2] Sustainable Building Design Lab, Dept. UEE, Faculty of Applied Sciences, University of Liege, Belgium
[3] Thermodynamics Laboratory, Aerospace and Mechanical Engineering Department, Faculty of Applied Sciences, University of Liège, Belgium
[4] Atmospheres and Monitoring lab (SAM), UR Spheres, University of Liege, Arlon Campus Environment, Avenue de Longwy 185, 6700 Arlon, Belgium.

*Correspondence to*: Sebastien Doutreloup (s.doutreloup@uliege.be)

**Abstract.** Increasing temperature due to global warming will influence building, heating and cooling practices. Therefore this dataset aims to provide formatted and adapted meteorological data for specific users who work in building designing,
architecture, building energy management systems, modeling renewable energy conversion systems, or other people interested in this kind of projected weather data. These meteorological data are produced from the regional climate MAR model simulations. This regional model adapted and validated over Belgium is forced firstly by the ERA5 reanalysis which represents the closest climate to reality, and secondly by 3 ESMs from the CMIP6 database, namely BCC-CSM2-MR, MPI-ESM.1.2, and MIROC6. The main advantage of using the MAR model is that the generated weather data have a high
resolution and are spatially and temporally homogeneous. The generated weather data follow two protocols. On one hand, the Typical Meteorological Year (TMY) and eXtreme Meteorological Year (XMY) files are generated following the method proposed by the standard ISO15927-4, allowing the reconstruction of typical and extreme years while keeping a plausible variability of the meteorological data. On the other hand, the HeatWaves Event (HWE) meteorological data are generated according to the method proposed by Ouzeau et al. (2016) used to detect the heatwave events and to classify them according
to three criteria of the heatwave (the most intense, the longest duration and the highest temperature). All generated weather data are freely available on the open-access repository Zenodo (https://doi.org/10.5281/zenodo.5606983 ; (Doutreloup, S. and Fettweis X., 2021).

**Keywords.** Climate simulation; Future weather files; Typical year; Extreme year; Heatwave; Regional Modeling; Belgium



## 1 Introduction

30       On a global scale, the warmest (SSP5-8.5) scenario from the IPCC last assessment report (IPCC et al., 2021) suggests a temperature increase of +5°C by 2100. However, the regions of the world will not warm up at the same speed or intensity. Some regions, such as the poles, will warm faster and larger than equatorial regions (Lee, J. Y. et al., 2021). Over the temperate oceanic regions such as Western Europe, the temperature is expected to increase between +1°C and +5°C in 2100 depending on the climate models and greenhouse gas emission scenarios used (Termonia et al., 2018; RMI, 2020;
IPCC et al., 2021).

Moreover, IPCC et al. (2021) affirm that extreme events will become more probable and more intense. In particular, the maximum temperature is expected to increase faster (sometimes up to twice) than the mean temperature (Seneviratne, S. I. et al., 2021). Over Western-Central Europe, maximum temperatures are projected to increase up to +7°C for a global increase of +5°C (Seneviratne, S. I. et al., 2021).

More concretely, during the summer season, hot extremes (including heatwaves) are already increasing and will continue to strengthen with global warming both in intensity and frequency (Suarez-Gutierrez et al., 2020; Seneviratne, S. I. et al., 2021; Dunn et al., 2020). The consequences of these heatwaves will affect human health (Fouillet et al., 2006), agriculture, the comfort and health of life inside buildings (Bruffaerts et al., 2018; Sherwood and Huber, 2010; Buysse et al., 2010), and the energy demand especially for cooling systems (Larsen et al., 2020). This is what motivated some previous studies in
Belgium to represent the energy needs for heating and cooling under average and extreme weather conditions (Ramon et al., 2019).

This comfort of life is precisely what motivates the ULiège OCCuPANt project (Impacts Of Climate Change on the indoor environmental and energy PerformAnce of buildiNgs in Belgium during summer, https://www.occupant.uliege.be/) in which climate data are involved. The OCCuPANt project aims to evaluate the vulnerability of building inhabitants in a warmer
climate context and more particularly during heatwaves.

The purpose of this dataset is to propose meteorological data coming from a fine resolution regional climate model over Belgium and the neighboring regions. The use of a regional model allows building spatially and temporally continuous and homogeneous past and future meteorological data according to different warming scenarios for some Belgian cities. This regional model is fed by the ERA-5 reanalysis model (Hersbach et al., 2020) to simulate the past climate (1980-2020) and
also by 3 different Earth System Models (ESMs) from the CMIP6 database (Wu et al., 2019; Tatebe et al., 2019; Gutjahr et al., 2019) to obtain different future projections and associated uncertainties for the same scenario SSP585.

For each city, considered period, model, and scenario, two synthetic files (in CSV format) are generated following ISO-15927-4: Typical Meteorological Year file (TMY) and eXtreme Meteorological Year file (XMY). In addition to these synthetic files, files focused on heatwaves are also generated, namely a file for the most intense heatwave event, one for the
warmest heatwave, one for the longest heatwave, and one containing all the heatwaves detected within a specific period. Finally, these files are described in detail in the methodology section.



## 2. Methodology

### 2.1 MAR model and area of interest

The regional climate model used in this study is the "Modèle Atmosphérique Régional" model (hereafter called "MAR") in
its version 3.11.4 (Kittel, 2021). The main role of MAR is to downscale a global model or reanalysis in order to get weather
outputs at a finer spatial and temporal resolution (Fig. 1). As shown in Fig. 1, MAR is a three-dimensional atmospheric
model coupled to a one-dimensional transfer scheme between the surface, vegetation, and atmosphere (Ridder and Gallée,
1998). Initially, the MAR model was developed for both Greenland (Fettweis et al., 2013) and Antarctica ice sheets (Agosta
et al., 2019; Kittel et al., 2021). However, it has recently successfully adapted to temperate regions such as Belgium
(Doutreloup et al., 2019; Wyard et al., 2021; Fettweis X. et al., 2017). In the framework of this study, MAR is initially
forced every 6 hours at its lateral boundaries (temperature, wind, and specific humidity) by the reanalysis ERA5 (called
hereafter MAR-ERA5) which is available at a horizontal resolution of ~31 km (Hersbach et al., 2020). Different kinds of
observations (*in situ* weather observation, radar data, satellites, *etc*) are 6-hourly assimilated into ERA5 to be the closest to
the observed climate. In this way, the simulations of MAR-ERA5 can be considered as a reconstruction simulation of the
current observed climate.

Then, the MAR model has been forced every 6 hours by 3 ESMs from the CMIP6 database (Eyring et al., 2016). These
ESMs do not contain any observational data and represent only the mean evolution of the climatic parameters. These models
contain two characteristic periods: one in the past, from 1980 to 2014 (hereafter called "historical" scenario), and another in
the future, from 2015 to 2100 according to different SSP scenarios (SSP585, SSP370, and SSP245). The selection and
description of each of these ESMs as well as a comparison with MAR-ERA5 over the historical period are presented in the
next section.

The atmospheric variables used to force MAR every 6 hours at each MAR vertical level are temperature, surface pressure,
wind, and specific humidity, as well as the sea surface temperature over the North Sea from both ERA5 reanalysis and the
three ESMs. The spatial resolution of MAR is 5km over an integration domain (120 x 90 grid cells) centered over Belgium
as shown in Fig. 2 to build hourly outputs.

The choice of 12 cities is motivated, on the one hand, by the size of the cities which must be sufficient to show a temperature
increase compared to the neighboring countryside and, on the other hand, to best represent climate spatial variability
observed in Belgium. For example, the city of Ostend is strongly influenced by the thermal inertia of the sea, while the city
of Arlon has a more continental climate.

### 2.2 Forcing models and MAR simulations

#### 2.2.1 Choice of representative ESMs





The Sixth Coupled Model Intercomparison Project (CMIP6; Eyring et al. (2016)) database contains about 30 ESMs from many scientific institutes around the world. For practical reasons, we can't regionalize all these ESMs. Thus, we had to select a few representative ESMs for our region of interest, West-Europe.

Our choice was based on two criteria. The first criterion is that the ESM should represent (with the lowest possible bias) the main atmospheric circulation in the free atmosphere over western Europe by evaluating the geopotential height at 500hPa and the temperature at 700hPa during summer and winter with respect to ERA5 over 1980-2014. After selecting the ESMs that meet this first criterion, the second criterion is to choose 3 ESMs representing the CMIP6 models spread in 2100 for the same scenario (SSP5-8.5 here). Namely, we keep only 3 ESMs: BCC-CSM2-MR (Wu et al., 2019), MPI-ESM.1.2 (Gutjahr

et al., 2019), and MIROC6 (Tatebe et al., 2019). The ESM BCC-CSM2-MR simulates warming close to the average of all 30 ESMs for the 2100 horizon with the SSP5-8.5 scenario, the ESM MIROC6 simulates larger warming than average, and the ESM MPI-ESM.1.2 simulates lower warming than average by 2100. The use of these 3 models allows us to obtain a first approximation of the uncertainty from ESMs without having to downscale all the 30 available models of CMIP6.

### 2.2.2 Future socioeconomic scenarios

Shared Socioeconomic Pathways (SSPs; Riahi et al. (2017)) are scenarios of global socio-economic evolution projected to 2100. These SSPs are used to develop greenhouse gas emission scenarios associated with different climate policies and are used to force each future ESMs. There are 3 main scenarios, namely SSP2-4.5, SSP3-7.0, and SSP5-8.5, which are respectively increasingly warming for 2100. For more details about these scenarios, we refer to Riahi et al. (2017).

For the same practical reasons that led us to choose only 3 ESMs out of the 30 available models, we cannot afford to

calculate every SSP scenario of each ESM. Thus, since the ESMs do not simulate general atmospheric circulation changes (Eyring, V. et al., 2021), only the SSP5-8.5 scenario has been calculated to save computational time. Then, scenarios SSP3-7.0 and SSP2-4.5 are reconstructed from the MAR simulations forced by the ESMs with the SSP5-8.5 scenario since the warning rates from the lower scenarios are included in this scenario SSP5-8.5 but for a different earlier period.

Therefore, this method involves determining the period of the SSP5-8.5 scenario that corresponds to the warming rate

projected by the other scenario for a given period. For this purpose, we use the raw data of each ESM and each scenario has been aggregated over Belgium to the horizon 2100. Then, for each ESM and for the SSP3-7.5 and SSP2-4.5 scenarios, we identify the corresponding period (the most equivalent mean and variability) in the SSP5-8.5 scenario for each decade. Then, we extract the weather data of this corresponding period for SSP3-7.5 and SSP2-4.5 out of the SSP5-8.5 forced MAR simulations (i.e., weather data of MAR-MPI with scenario SSP5-8.5 during the period 2066-2085 represent MAR-MPI with

scenario SSP3-7.0 for the period 2081-2100).



### 2.2.3 Evaluation of the MAR simulations

To verify that these ESMs forced MAR simulations can be used to anticipate future periods, it is necessary to evaluate them over the present period (namely 1980-2014 which corresponds to the period of the historical scenario). The aim is to determine if they are able to represent the average and the climate variability over the Belgian territory for this period as

observed (i.e. ERA5 in your case). So, we compare the 3 MAR simulations forced by the 3 ESMs with the MAR simulation forced by the ERA5 reanalyses. In this case, as the two most important variables for this database are the temperature at 2m a.g.l. and the incident solar radiation, the means and standard deviation of these data over the period 1980-2014 and over the Belgian territory are compared in Tab. 1 and Fig. 3. These values are compared on an annual scale as well as on a summer scale since the OCCuPANt project focuses on heatwaves.

The results of this comparison presented in Table 1 indicate that the incident solar radiations and temperature at 2m a.g.l. values proposed by the 3 MAR simulations forced by the ESMs are very close to the average simulated by MAR-ERA5 with (not statistically significant) biases between MAR forced by the ESMs and MAR-ERA5 lower than the standard deviation (i.e. the interannual variability) of MAR-ERA5. We can also note that the interannual variability of the MAR simulations forced by the ESMs is close to the interannual variability of the MAR-ERA5 simulation. We can then conclude that the

MAR simulations forced by the ESMs are able to represent the current climate and its interannual variability with success, except MAR-MIR which significantly overestimates temperature and solar radiation in summer. Knowing that MAR-MIR simulated the largest warming in 2100, this simulation needs to be considered as the extreme climates we could have.

### 2.3 Generating the TMY and XMY files

The Typical Meteorological Year (TMY) and the eXtreme Meteorological Year (XMY) are datasets that are widely used by

building designers and others for modeling renewable energy conversion systems (Wilcox and Marion, 2008). The TMYs are the synthetic years (on an hourly basis) constructed by representative typical months (Barnaby and Crawley, 2011) which are selected by comparing the distribution of each month within the long-term (minimum 10 years) distribution of that month for the available observations or modeled data (using Finkelstein-Schafer statistics (Finkelstein and Schafer, 1971). The XMY is the extension of the TMY weather data and is formed by selecting the most deviating (i.e., extreme) months over a

certain dataset instead of typical months (Ferrari and Lee, 2008). There are many methods to reconstruct this kind of weather file (Ramon et al., 2019), but for the purpose of this study, a protocol for the construction of these typical years has been developed based on the ISO15927-4 (European Standard, 2005) and is described briefly below.

The method consists of reconstituting each month of the typical (resp. extreme) year with the most typical (resp. extreme) month present in the considered period for a considered city. The comparison is essentially based on two variables, namely

temperature at 2m a.g.l. and incident solar radiation. The choice of these two climatic parameters (called "*para*" hereafter) is related to the fact that they both influence the comfort inside the buildings. Therefore, we generate files with the typical year according to the temperature at 2m a.g.l. and, a typical year according to the incident solar radiation.



Here are the steps to find the most typical (resp. extreme) month :

1  From all the hourly data of all the same calendar months available within the selected period, the daily mean of
*para* is calculated;

2  For each calendar month, the percentile 50 (resp. 95) of *para* is calculated over the period covered to find the month
which is the closest to the 50 percentile (resp. 95) of *para*;

3  Finally, the hourly weather values of this typical (resp. extreme) month are stored in the file of the typical (resp.
extreme) year.

The hourly weather variables available in the TMY and XMY files are:

1  Dry bulb temperature at 2m a.g.l. (°C)

2  Relative humidity at 2m a.g.l. (%)

3  Global horizontal radiation (Wh/m²)

4  Diffuse solar radiation (Wh/m²)

5  Direct normal radiation (Wh/m²)

6  Wind speed at 10m a.g.l. (m/s)

7  Wind direction (north degrees)

8  Dew point temperature at 2m a.g.l. (°C)

9  Atmospheric Pressure (Pa)

10  Cloudiness (in tenths)

11  Sky temperature (K) according to (Duffie and Beckman, 2013)

12  Specific humidity at 2m a.g.l. (kg/kg)

13  Precipitation (mm)

**2.4 Definition of a heatwave and generating the HWE files**

In Belgium, heatwaves are officially defined by two definitions, a retrospective one, and a prospective one. The retrospective
heatwave is defined as periods of at least 5 consecutive days with a maximum temperature higher than 25°C, of which at
least 3 days within this period with a maximum temperature higher than 30°C (RMI, 2020). The prospective heatwave is a
period with a predicted minimum temperature of 18.2°C or more and a maximum temperature of 29.6°C or higher both for 3
consecutive days (Tsachoua, L. and Reynders, D., 2016).

However, these definitions are static and do not consider the local climate of each region. For example, on the highlands
where it is on average colder, these heatwave criteria are not necessarily met, even though this region also experiences a
heatwave. Moreover, when comparing the different ESMs, this fixed heatwave definition criterion could induce artifacts
since each ESM has its own variability and biases over the current climate. For these reasons, we have used the statistical





definition of a heatwave from (Ouzeau et al., 2016) computed for each MAR pixel regardless of its basic climate, and each
185  ESM independently of its own internal variability.

The calculation method, according to (Ouzeau et al., 2016), is as follows and it is illustrated by Fig. 4:

▪  For the period 1980-2014 (which corresponds to the "historical" scenario in the ESMs), for each pixel and each MAR simulation, we calculate three thresholds defined by 3 percentiles of the daily mean temperature: Sint = 95th percentile, Sdeb = 97.5th percentile and Spic = 99.5th percentile ;

190  ▪  A heatwave is detected when the daily mean temperature reaches Spic. The duration of this event is the number of days between the first day when the daily mean temperature is equal to or greater than Sdeb and either when the daily mean temperature falls below Sint or when the daily mean temperature falls below Sdeb for three consecutive days at least.

▪  In this dataset, we add a condition compared to (Ouzeau et al., 2016) which is that the minimum duration of the heatwave must be at least 5 consecutive days otherwise the heatwave event is not considered.

195  Once the heatwaves events (called "HWE" hereafter) are detected, we can characterize them according to three criteria:

1  the duration which is the number of consecutive days of the HWE;

2  the maximal daily mean temperature reached during the HWE;

3  the global intensity which is calculated by the cumulative difference between the temperature and the Sdeb threshold during the HWE, divided by the difference between Spic and Sdeb.

The hourly data provided in each HWE file are the same as for the TMY and XMY files (see section above). For each period, each city, each scenario, and each forcing, 4 files are created:

▪  a file containing hourly weather data for the longest HWE;

▪  a file containing the hourly weather data of the HWE characterized by the highest daily average temperature;

▪  a file containing the hourly weather data of the HWE characterized by the highest intensity;

▪  a file that concatenates all the hourly weather data of all the HWEs present in a period, for each city, each scenario, and each MAR model.

## 3. The dataset

### 3.1 Data availability

The data described in this manuscript can be freely accessed on the Zenodo open-access repository:
https://doi.org/10.5281/zenodo.5606983 (Doutreloup, S. and Fettweis X., 2021). As the files are numerous, each zipped folder contains all the data for each city concerned by this dataset.

### 3.2 Structure of each file

All files are in CSV format comma-separated. The file names are formatted in such a way that they contain all the information about the origin of the file. Each file name is composed as follows:



-       For TMY or XMY: CityName_Type_Period_Scenario_MARmodel_Para.csv

    -       For Heatwave: CityName_Type_Period_Scenario_MARmodel.csv

with :

-       CityName: the name of the city considered in this file;

-       Type: the nature of this file, namely TMY (typical meteorological year), XMY (extreme meteorological year),

HWE-HI (most intense heatwave event), HWE-HT (warmest heatwave event), HWE-LD (longest heatwave event), and HWE-all (all heatwave events);

-       Period: the time period considered;

-       Scenario: the IPCC scenario of the ESM considered, namely "hist", "SSP5-8.5", "SSP3-7.0", and "SSP2-4.5" (note that for MAR-ERA5 the name of the scenario is set by default to "hist" even if the ERA5 forcing does not contain any

scenario);

-       MAR model: the version of MAR used, namely: MAR-ERA5, MAR-BCC, MAR-MPI or MAR-MIR;

-       Para: is the climatic parameter used to generate the TMY or XMY file, namely the temperature at 2m a.g.l. (TT) or the incident solar radiation (SWD).

For TMY and XMY files, the header is composed of the task number (which is a reference number for internal use in the

OCCuPANt project), the name of the city, and the name of the different variables accompanied by their unit. For HWE files, the header is composed of the task number (same remark as above), the name of the city, the characterization of the heatwave (longest, warmest, and most intense) accompanied by its period, and the name of the different variables accompanied by their unit. Then comes the weather data with the time variable in the first column. Weather data are in universal time and leap year mode for the MAR-ERA5 and in no-leap year mode for MAR-BCC, MAR-MPI, and MAR-

MIR.

A short example of how to select the desired data and files, as well as an example of how to use it, is provided in Appendix A for Liege-City.

## 4. Conclusion

The goal of this dataset is to provide formatted and adapted meteorological data for specific users who work in building

designing, architecture, building energy management systems, modeling renewable energy conversion systems, or other people interested in this kind of data. These weather data are derived from the regional climate MAR model. This regional model adapted and validated over Belgium is forced by 4 ESMs. On one hand, MAR is forced by the ERA5 reanalysis to represent the closest climate to reality. On the other hand, MAR is forced by 3 ESMs from the CMIP6 database, namely BCC-CSM2-MR, MPI-ESM.1.2, and MIROC6. The main advantage of using the MAR model is that the generated weather

data at high resolution are spatially and temporally homogeneous.





The generated weather data follow two protocols. Firstly, the TMY and XMY meteorological data are generated according to the method proposed by the standard ISO15927-4, allowing the reconstruction of typical and extreme years while keeping the plausible variability of the meteorological data. Secondly, the meteorological data concerning heatwaves (HWE) are
generated according to the method proposed by Ouzeau et al. (2016) to detect the heatwave events and to classify them according to three criteria of the heatwave. The OCCuPANt project, in which this paper is included, aims to identify the highest temperature, the longest heatwave duration, and the most intense heatwave event. Finally, all generated weather data are open source and freely available on the open repository Zenodo (https://doi.org/10.5281/zenodo.5606983 ; (Doutreloup, S. and Fettweis X., 2021).

**5 Appendices**

**5.1 Appendix A: Example for Liege city**

The Liege folder contains 596 files. This is a huge amount of files, so to find your way around we suggest the following method for TMY and XMY files :

1        Choosing a typical or extreme year, i.e. select XMY for an extreme year;

2        Choosing the parameter that determines the typical or extreme year, i.e. if you want a year with extreme incident solar radiation (i.e. sunshine), you should select SWDbased.

3        Choosing the reference period, i.e. 2085-2100 for the end of the century;

4        Choosing the socio-economic scenario, i.e. "ssp585" for a world that continues to use fossil fuels;

5        Choosing one of the 3 MAR models, i.e. MAR-BCC for the MAR model forced by BCC-CSM2-MR;

6        Finally, the choice is made for the file:

 Liege-City_XMY2085-2100_ssp585_MAR-BCC_SWDbased.csv

This file selection method is a proposal but each user is free to develop his own method and of course you can use several files to compare them. For example, if we want to compare typical and extreme temperatures for the end of the century and for the two selected parameters with MAR-BCC, we get Fig. A-1. Fig. A-1 clearly shows that the extreme year temperature
based on temperature is much higher than the typical year temperature. On the other hand, the extreme year temperature based on incident solar radiation is much lower, especially in January, than the typical year temperature because extreme radiation in winter usually means cold weather.

The method is almost identical for the heatwave files except that there are no parameters to determine the typical and extreme years but we have to choose if we want to look at the longest, the most intense, the hottest, or all heatwaves present
within the period. Therefore, if we want to compare the longest heatwave (HWE-LD) over the same period, for the same model and scenario as the example above, we choose the file :

Liege-City_HWE-LD_2085-2100_ssp585_MAR-BCC.csv

Fig. A-2 compares the temperature evolution during the warmest heatwave obtained from each of the MAR-BCC, MAR-
MPI, and MAR-MIR simulations. The header of each file shows the warmest daily average temperature, namely 37.2°C for
MAR-BCC, 35.3°C for MAR-MPI, and 37.8°C for MAR-MIR. However, it can be seen in Fig. A-2 that the hourly
temperatures obviously rise much higher than these daily average values and in this case rise to ~44°C for MAR-BCC and
MAR-MIR and ~42°C for MAR-MPI.

## 6 Author contribution

All authors participated in the conceptualization of this paper. SD and XF designed the experiments and methodology and
SD carried them out. SD prepared and created figures and data. XF oversaw the research. SD wrote the initial manuscript
and all authors participated in the revision.

## 7 Competing interests

The authors declare that they have no conflict of interest.

## 8 Acknowledgment

This research was partially funded by the Walloon Region under the call "Actions de Recherche Concertées 2019 (ARC)"
and the project OCCuPANt, on the Impacts Of Climate Change on the indoor environmental and energy PerformAnce of
buildiNgs in Belgium during summer. The authors would like to gratefully acknowledge the Walloon Region and Liege
University for funding. Computational resources have been provided by the Consortium des Équipements de Calcul Intensif
(CÉCI), funded by the Fonds de la Recherche Scientifique de Belgique (F.R.S.-FNRS) under Grant No. 2.5020.11 and by the
Walloon Region.

## 9 Financial support

This research was partially funded by Wallonia-Brussels Federation grant for Concerted Research Actions 2019 (ARC) and
the project OCCuPANt, on the Impacts Of Climate Change on the indoor environmental and energy PerformAnce of
buildiNgs in Belgium during summer.

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





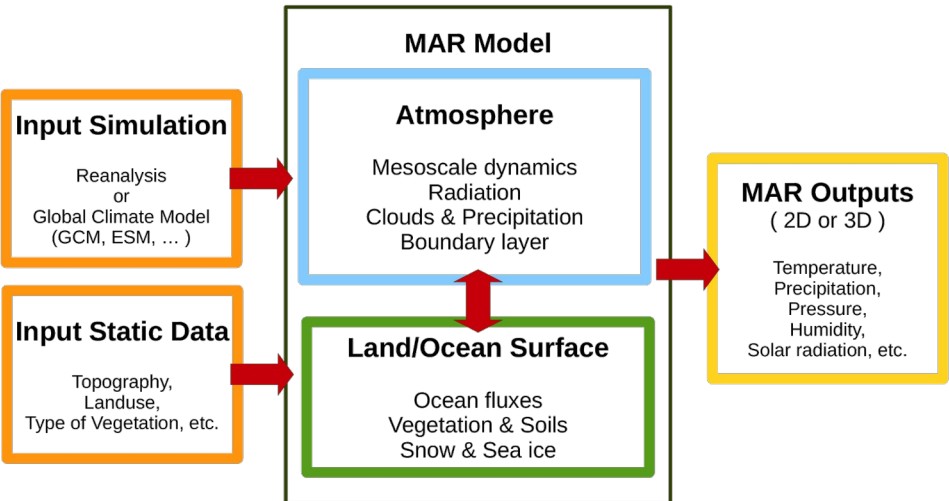


**Figure 1: Workflow of the MAR model. The MAR model (black box) needs to be forced by a reanalysis model or global model (upper orange box) and by static data (lower orange box) and the result of the MAR simulation gives meteorological variables in two or three dimensions (yellow box).**



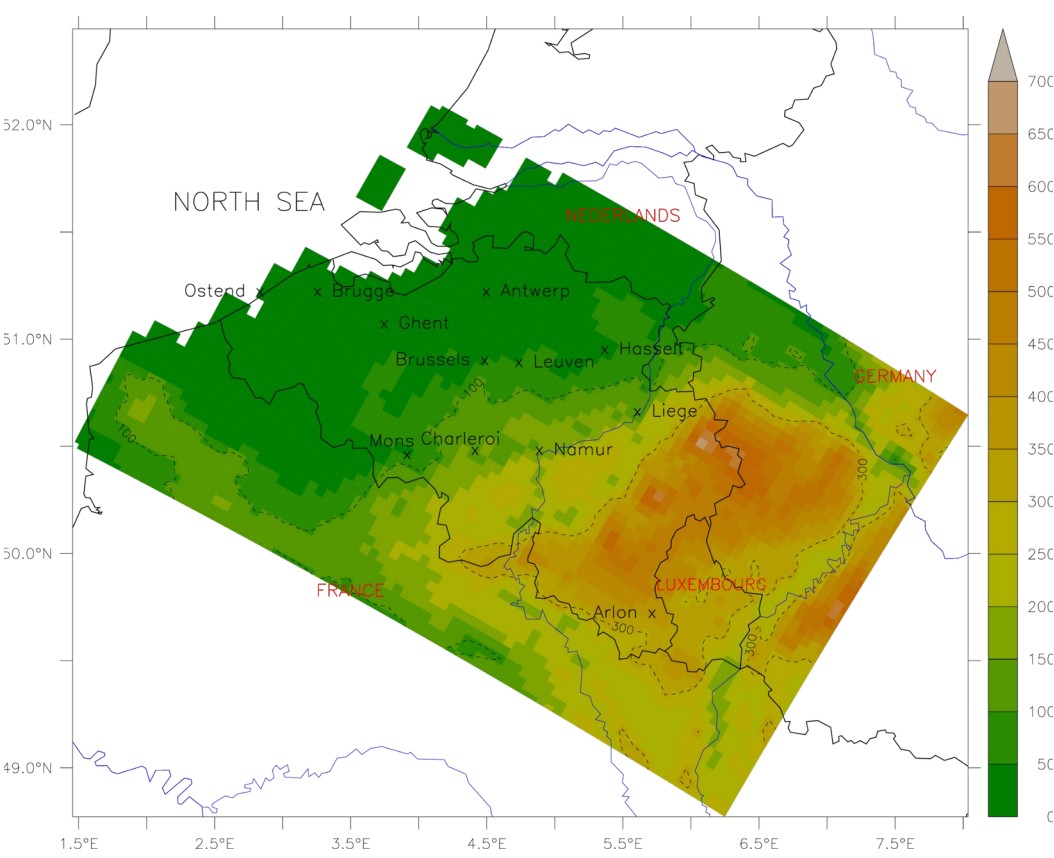

**Figure 2: Model topography (color background, in meters), localization of cities (black) used in this dataset for Belgium, and**
**localization of neighboring countries (in red).**





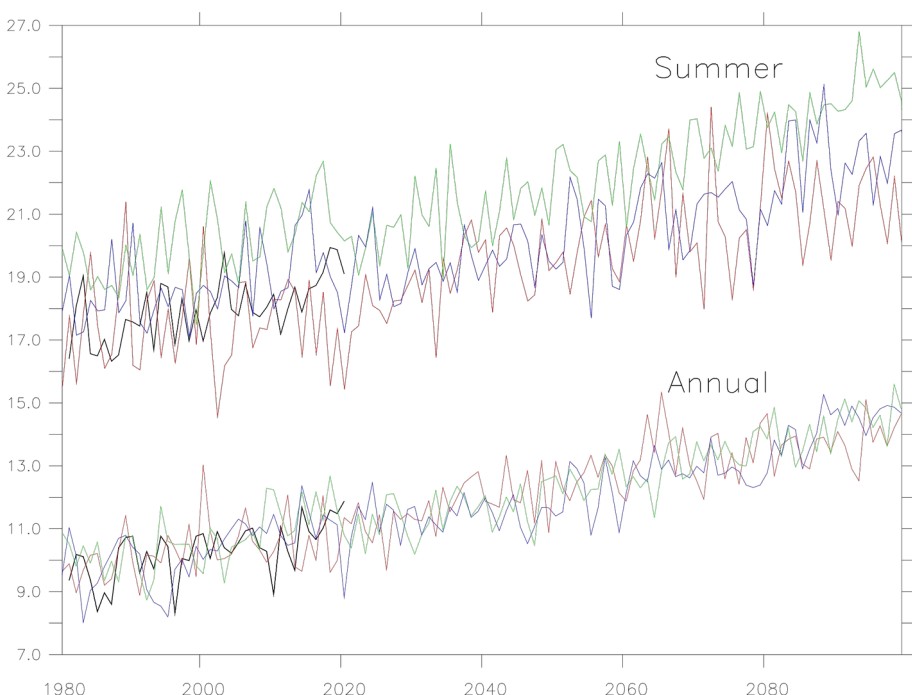

**Figure 3: Annual mean temperature (in C, lower lines) and annual summer temperature (in C, upper lines) of MAR forced by ERA5 reanalysis (in black bold lines), MAR forced by BCC-CSM2-MR (in red lines), MAR forced by MPI-ESM.1.2 (in blue lines), and MAR forced by MIROC6 (in green lines) between 1980 and 2100 according to the scenario SSP5-8.5. The average is computed here over the whole integration domain excluding the ocean.**

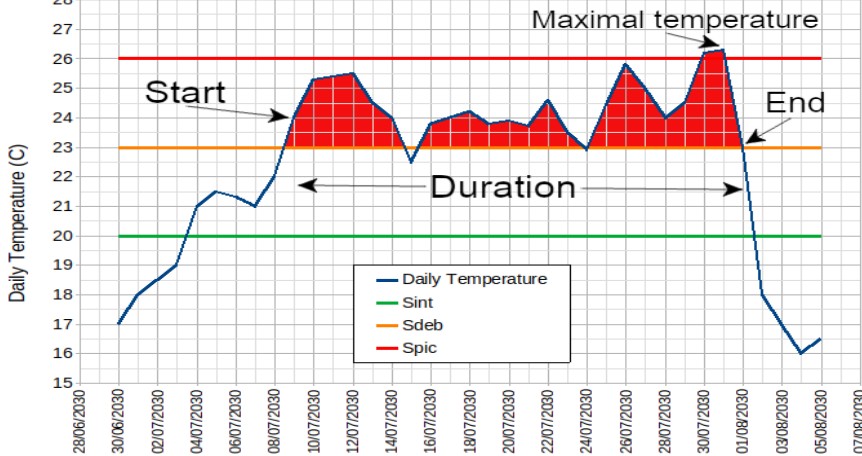

**Figure 4: Characterization of a fictional heatwave from a daily mean temperature indicator (example of a fictional time series from the 30th June 2030 to 5th August 2030): Duration (start and end), maximal temperature, and global intensity (red area of the plot). This figure and caption are inspired from Ouzeau et al. (2016).**



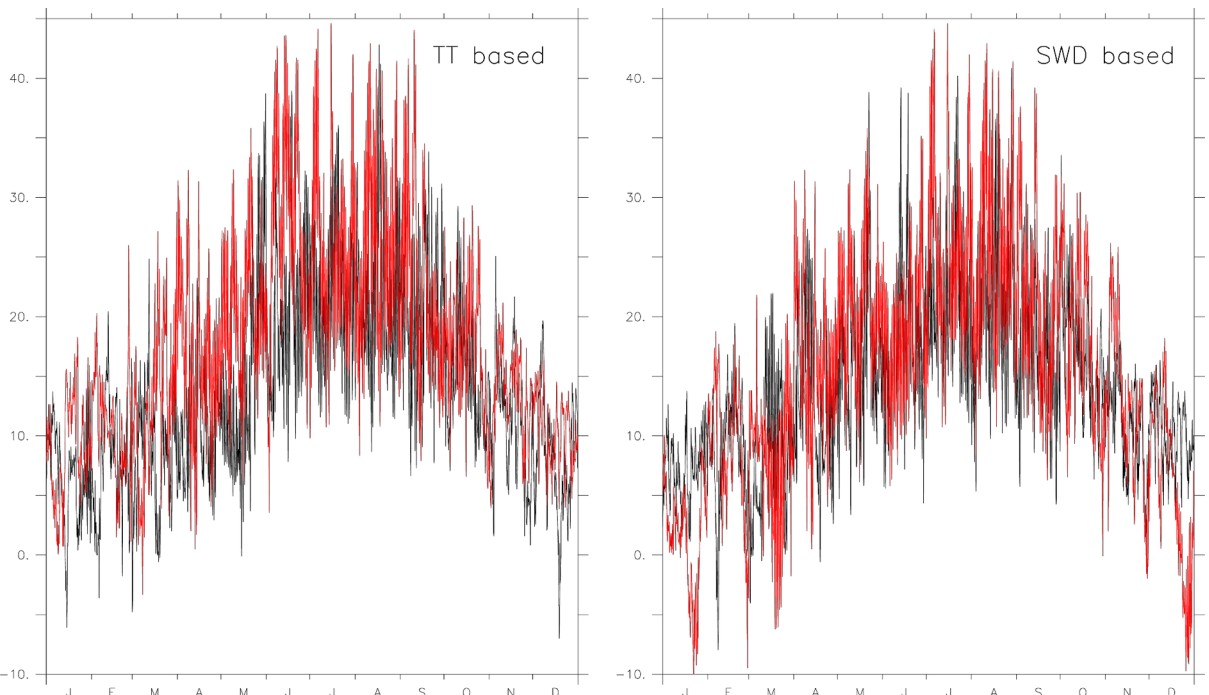

**Figure A-1: Typical (black) and extreme (red) temperature for Liege city simulated by MAR forced by BCC-CSM2-MR with the scenario ssp585 for the period 2085-2100. The typical/extreme months are based on temperature (left) and incident solar radiation**
**(right). These data are extracted from these files :**
- **Liege-City_TMY2085-2100_ssp585_MAR-BCC_TTbased.csv**
- **Liege-City_XMY2085-2100_ssp585_MAR-BCC_TTbased.csv**
- **Liege-City_TMY2085-2100_ssp585_MAR-BCC_SWDbased.csv**
- **Liege-City_XMY2085-2100_ssp585_MAR-BCC_SWDbased.csv**

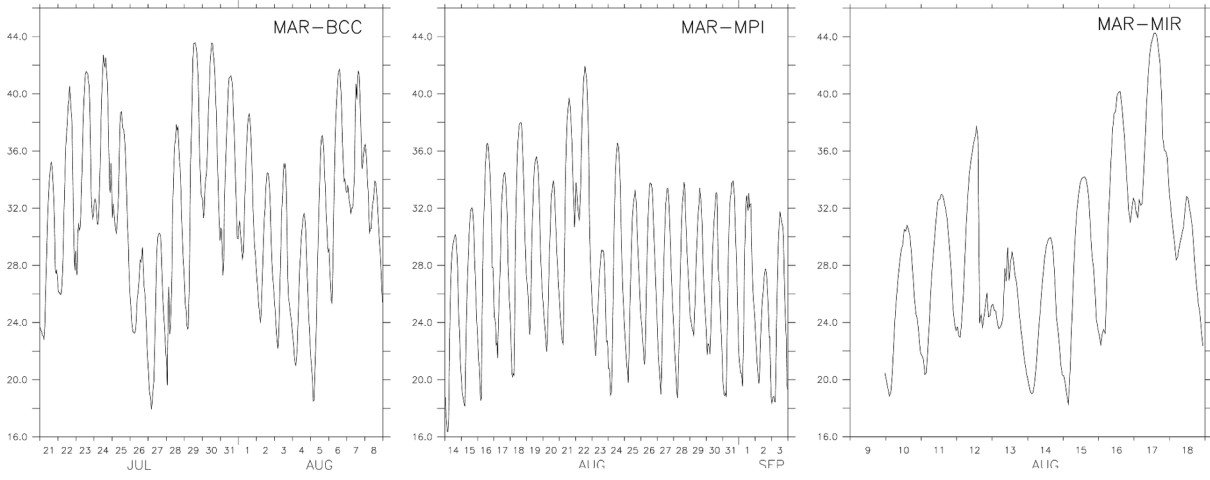


**Figure A-2: Temperature during warmest heatwave events for Liege city simulated by MAR-BCC (i.e., MAR forced by BCC-CSM2-MR), MAR-MPI (i.e., MAR forced by MPI-ESM.1.2), and MAR-MIR (i.e., MAR forced by MIROC6) with the scenario ssp585 for the period 2085-2100. These data are extracted from these files :**
- **Liege-City_HWE-HT_2085-2100_ssp585_MAR-BCC.csv**





- **Liege-City_HWE-HT_2085-2100_ssp585_MAR-MPI.csv**
    - **Liege-City_HWE-HT_2085-2100_ssp585_MAR-MIR.csv**

| 1980-2014 over Belgium territory | Time scale | MAR-ERA5 | MAR-BCC | MAR-MPI | MAR-MIR |
|---|---|---|---|---|---|
| *Mean annual temperature at 2m a.g.l. (in C)* | *annual* | 10.1 +/- 6.2 | 10.3 +/- 6.0 | 10.1 +/- 6.4 | 10.4 +/-7.2 |
| | *summer* | 17.7 +/- 1.3 | 17.6 +/- 1.7 | 18.6 +/- 1.2 | 19.9 +/- 1.1 |
| *Mean incident solar radiation (in W/m²)* | *annual* | 119 +/- 78 | 122 +/- 79 | 116 +/- 83 | 134 +/- 85 |
| | *summer* | 213 +/- 20 | 221 +/- 20 | 224 +/- 22 | 239 +/- 25 |

**Table 1: Comparison of MAR-ERA5 (i.e., MAR forced by the reanalysis ERA5) with MAR-BCC (i.e., MAR forced by BCC-CSM2-MR), MAR-MPI (i.e., MAR forced by MPI-ESM.1.2), and MAR-MIR (i.e., MAR forced by MIROC6) to mean temperature at 2m a.g.l. and mean incident solar radiation over Belgium territory during 1980-2014.**