# Peer review of "Historical and Future Weather Data for Dynamic Building Simulations in Belgium using the regional climate model MAR: Typical & Extreme Meteorological Year and Heatwaves"

_Earth System Science Data, 2021_

## Author Comment (AC1)

**RC1: 'Comment on essd-2021-401', Anonymous Referee #1, 29 Mar 2022**

General comments

The article is appropriate to support the publication of the data set. It both build on previous work and provides new approaches to producing future weather data files that can be used by a wide range of end users. The data appear to be of quality, complete and usefull.
As the construction of future weather data files is not a trivial task, most of my comments ask for clarification of certain methodological choices and assumptions. I leave it to you to modify the text accordingly if you deem it necessary to enable end-users to better understand the data they are using.

*First of all, we would like to thank reviewer #1 for his/her useful and constructive remarks. Our answers to the reviewer comments are in blue below.*

Specific comments

Abstract

Maybe add a short sentence explaining that this work is part of the ULiège OCCuPANt project (it might help future readers find the article and the data).

*It's added at the end of the abstract : « [...] and these data are produced within the framework of the research project OCCuPANt (https://www.occupant.uliege.be/ - ULiège). »*

2.1 MAR model and area of interest

Line 74. "can be considered as a reconstruction simulation", I understand what you mean but let's keep in mind that the MAR model is not necessarily perfect and so the MAR-ERA5 can still be biased. I come back to this point in the 2.2.3 Evaluation section.
by
*We changed this sentence to avoid any ambiguity :*
*« In this way, the simulations of MAR-ERA5 can be considered as the closest simulation to the current observed climate. »*

Line 77. "represent only the mean evolution", I understand what you want to say but I would suggest to reformulate the sentence to make it clearer that we are not just talking about the average climate but also its variability (and especially extremes), otherwise you would not be able to use these simulations to produce the XMY files.

*We changed this sentence to:*
*« These ESMs do not contain any observational data and represent only an evolution of the average and interannual variability of the climatic parameters over the long period. ».*

Line 84. What do you mean by "integration domain"? As I understand it is the final domain where data can be extracted and where points from the "relaxation zone" are omitted?

*The reviewer is right. We have added this sentence to precise what we mean by the integration (i.e. the simulated area without the pixels impacted by the relaxation zone).*

Line 85. "The spatial resolution of MAR is 5km". Do you directly downscale ERA5 and GCMs simulations to 5km or do you make intermediate lower resolution simulations to reduce the "resolution jump" between the two?

We directly downscaled ERA5 and GCMs simulations to 5km. This is a question which comes up regularly. Previous works (including same references as in Section 2,1) demonstrated that the MAR model produces better results with a direct downscale knowing that an intermediate lower resolution increases the MAR biases (due to an amplification of biases).

Lines 83-84. I would suggest adding the different period you simulated: "[…] over the North Sea from both ERA5 reanalysis for the YYYY-YYYY period and the YYYY-YYYY period for the three ESMs". I think it would make it easier to understand some parts of the next section.

We have precise this passage in this way.

Lines 86-87. You indicate that your choice of cities is based on the Urban Heat Island (UHI) effect ("show a temperature increase compared to the neighbouring countryside") but I do not think the MAR model incorporate any sort of Urban Canopy Models to represent cities. Are you using another approach on urban points (replacing them by rock for example)?
Depending on the approach you are using some points/clarification need to be clear (not necessarily in the article but for the people using the data afterwards):
In the case you were using rocks, depending on the physical properties associated to this cover I think we could expect an underestimation of nocturnal temperatures and of the UHI. Consequently if we were to compare the data from the MAR-ERA5 simulation to observations or even to ERA5 we might find different UHI patterns/intensities.
In the case you were not representing cities at all (which is fine in my opinion), users need to know that the data they are using correspond to the rural conditions surrounding the city.

The MAR model simulates urban heat islands through a modification of both the surface albedo which is fixed to 0.1 and by the absence of vegetation which modifies the thermal and humidity exchanges with the atmosphere compared to the surrounding countryside. Thus, it is not a model of urban canopy *stricto sensus* but we approach this way with a « low-cost method ». We have added a sentence at the beginning of chapter 2.1 to clarify this:  « *The MAR model also includes an urban island module which modifies the city grid points to simulate urban heat island through a modification of the surface albedo (fixed to 0.1) and an absence of vegetation which influence the thermal and humidity exchanges between soil and atmosphere.* »

2.2.1 Choice of representative ESMs

Line 95. Just by curiosity/for clarification. How is your first criterion defined to preselect the GCMs with the lowest possible bias relative to ERA5? How do you combine the geopotential height and temperature biases? How do you select GCMs based on these biases and how much GCMs do you retain after this first selection?
Once you've made your first selection did you look at the spread of your sub-ensemble compared to the 30 GCMs? On both the historical temperature as well as the expected warming rate? My main concerns is that some GCMs could perform well in reproducing the historical climate (at least in terms of atmospheric circulation) but produce unrealistic warming (some CMIP6 models have been shown to be outside the IPCC's "likely" range of expected warming; Zelinka et al., 2020).

The selection of these 3 GCMs was made after evaluation of their abilities to represent the current (1976-2005) average climate over Europe with respect to ERA5, on the basis of the skill score

methodology developed by Connolley and Bracegirdle (**2007,** 10.1029/2007GL031648). We have now precised this in the manuscript. But the reviewer is absolutely right, GCMs that work well on the present could completely diverge in the future. But this is not the case for the 3 GCMs selected here. In the future, we plan to publish on Zenodo other datasets produced by other forcing GCMs. This will improve the confidence interval in our climate projections.

Line 98. When you talk about "the models spread in 2100", are you comparing end of the century temperatures directly (between models) or warming relative to the historical period? Because later in the paragraph the formulation change to "warming". For example, when you states that "MIROC6 simulates larger warming than average" do you talk about warming relative to the historical period or just that it simulates higher temperatures than the average (in which case it could be either a greater warming rate, a warm bias or a combination of both).

We first calculated the average warming at the end of the century by using 30 GCMs. We then compared the best GCMs with respect to the average warming of the 30 GCMs by selecting a warmer, a colder and a mean GCM compared to the ensemble mean warming rate. We have a bit improve our sentence : « *The ESM BCC-CSM2-MR simulates a warming close to the ensemble mean of the 30 ESMs for the 2100 horizon using the SSP5-8.5 scenario, the ESM MIROC6 simulates a larger warming than the ensemble mean, and the ESM MPI-ESM.1.2 simulates a lower warming than the ensemble mean by 2100.* »

2.2.2 Future socioeconomic scenarios

Lines 117-120. Some things are not clear to me on your approach to chose a period representative of others SSPs warming at the end of the century in the SSP5-8.5 GCMs. If I understood correctly you compute the projected average temperature over the whole of Belgium at the end of the century in each ESM and for the SSP3-7.5 and SSP2-4.5 scenarios (on which period? For all periods used in the dataset)
and then you search for the "closest" decade in the SSP5-8.5 in terms of average temperature and variability (how do you define it? With the help of the standard error),
lastly you extract the period you found in your MAR simulations. On line 117 you say you search for a "decade" (do you compute a moving average or do you just take every 10 years No, this is a mistake, we have deleted « for each decade » in our manuscript, this is for each period).
Also the example you give afterwards show a 20 years period. Lastly, just to be sure, do you apply this approach on the 3 ESMs you have previously selected or on the whole ensemble?
This method is apply for each the 3 selected ESMs.
This section, and especially the approach to "reconstruct" different SSPs from a single one is interesting but I would be cautious on multiple points.
We are aware that this method is open to discussion. However, it allows us to have results as a first approximation without addition computer time.

Lines 111-112. "scenarios SSP3-7.0 and SSP2-4.5 are reconstructed from the MAR simulations forced by the ESMs with the SSP5-8.5 scenario". Are you aware of previous work that has implemented a similar approach?
Even if the end-of-century warming level of all scenarios is found in the SSP5-8.5, they are probably not perfectly comparable: for example one can expect to see different local conditions between a world that gradually reaches 2°C of warming in 100 years and a world that reaches it in a few decades.
We are not aware of any previous work that has done about this. However, our methodology has been inspired by the work of the IPCC, especially when it develops the "global warming levels" +1,5°C, +4°C, etc. Indeed, this methodology presented in AR6 allows for example to compare easily and quickly the projections made for AR5 (using RCPs) with AR6 (using SSP). In both cases

(the IPCC way and ours), the method is questionable since the climate does not respond linearly to different GHG scenarios as the warming rates (as the reviewer indicates) are not comparable, the Earth-Atmosphere-Ocean system does not respond in the same way. However this methodology allows us to obtain a first approximation of the climate changes according to the studied scenarios, but we must be careful about the interpretation of the results. That is why, we added a sentence to alert the reader and the user of this data:

*"This method is open to discussion for several reasons, firstly because climate does not react in a linear way to an increase of GHG flowing in different SSP scenarios. Moreover, with an equal warming rate but different periods, the Earth, Atmosphere and Ocean systems will not have the same (spatial and temporal) responses due to their inertia. Despite these precautions, the method used here allows on the one hand a very quick estimation without additional computer time. On the other hand, it remains valid as a first approximation, especially since the most interesting weather variable in this study is temperature, which is, therefore, the least sensitive to these issues."*

Therefore, when sharing the data, I would make it clear that the scenarios other than the SSP5-8.5 do not really correspond to GCM driven by these scenarios but are reconstructed based on warming level. And for future work, if the same approach is applied, I would suggest to think about presenting the data not in terms of SSPs scenarios but in terms of expected warming level (both global and European).

Thanks a lot for this recommendation. This is an excellent idea keeping in mind that the same problem as the remark above will also arise.

I come back to another point in the next section 2.2.3. regarding known differences in warming simulated by RCMs and their driving GCMs – which I expect could also be found for MAR simulations.

2.2.3. Evaluation of the MAR simulation

Line 131. "are very close the average" is debatable with MAR-MIR being more than 2°C hotter on average in summer.

It's true but taking into account the interannual variability, we can consider that this is not significantly different than the average. This is explained in the second part of this sentence.

Line 132. "not statistically significant". How do you compare both average and how do you test for statistical significance?

We consider that the difference between two MAR simulations is not statistically significant when the difference between the mean of MAR-(ESM) and MAR-ERA5 is less than the standard deviation (i.e. interannual variability) of MAR-ERA5.

Line 132. "lower than the standard deviation of MAR-ERA5". Not true for MAR-MIR in summer for temperature.

Indeed! Thanks. We corrected it by "[…] are mostly close […]" without precision because we discuss the case of MAR-MIR a few sentences later.

Lines 131-134. I would be more cautious about the conclusions.
Since the ESMs driven MAR simulations are compared to another MAR simulation you can not evaluate the intrinsic biases of MAR (which has been shown to have a slight warm bias (between

+0,3 and +0,7 °C on average annually with a negative bias in winter and a positive bias in summer) by Wyard et al., 2017 [using an older version]). A stricter evaluation of MAR would have been to compare the MAR-ERA5 simulation to observations.

We have adapted this sentence by being more careful : "We can then conclude that the MAR simulations forced by the ESMs are able to represent the mean climate simulated by MAR-ERA5 […]"
However, as the reviewer mentioned, the version of MAR between Wyard et al.(2017) and this paper is not the same, some significant improvements and biases corrections of the model have been made from this time. Another paper is currently under review in which we compare MAR-ERA5 simulations over Belgium and northern France with observed data. This paper shows that the annual bias has been reduced as well as the positive and negative seasonal biases over winter and summer compared to Wyard et al (2017).

"expect MAR-MIR which significantly overestimates […]". Here you correct your prior statements made on lines 131-132.

Lines 131-132 have been corrected.

"Knowing that MAR-MIR simulated the largest warming in 2100 […]". This point makes me come backs to the questions I asked regarding the way you define warming in the section 2.2.1: is MIROC6 really the ESM with the largest warming relative to the historical period or is it just a more "warmly-biased" ESM than the others?

This is a more warmly-biased ESM than the others we have selected according to the skill score method (i.e. represent the main atmospheric circulation over Western Europe). It is not the ESM with the largest warming: UKESM1-0-LL or CESM2 are more climate sensitive for example.

As I briefly mentioned in the previous section, previous works have shown that some regional climate models are not able to reproduce the warming of their driving GCMs/ESMs over Europe (Schwingshackl et al., 2019; Boé et al., 2020). Different hypothesis have been put forward such as the lack of evolution of aerosols (Boé et al., 2020; Gutiérrez et al., 2020) or the physiological effect of CO2 (Schwingshackl et al., 2019; Boé et al., 2019). See Ribes et al., 2022 for a comparison of CMIP6, CMIP5 and EURO-CORDEX warming over France (which might be comparable to Belgium). I was wondering if you compared the warming simulated by the ESMs to the one simulated by MAR?

Below, there is the warming rate as simulated by MAR and by the driving. It is the slope of the linear regression (in °C/month) using monthly temperatures over 1980-2100:

|  | BCC | MPI | MIR |
| --- | --- | --- | --- |
| ESM | +0,00365 | +0,00342 | +0,00364 |
| MAR-ESM | +0,00362 | +0,00338 | +0,00363 |

Following this point and the one I made previously regarding the use of SSPs names when sharing the data, it might be interesting for future work to build files on projected warming instead of SSPs. I stress these points because each new step in the cascading sources of uncertainties from SSPs, ESMs, RCMs and finally their application (with building energy models for example) makes it extremely difficult in the end to estimate the confidence we can have in the final results.

Thanks for this suggestion.

2.3 Generatin the TMY and XMY files

Lines 154-159. This section might need to be reformulated. I don't think I understood the different steps correctly. As an example para is the temperature and the period is 2001-2010:
(1) You take all the hourly data of all the months of each month one by one (January 2001, February 2001, …, November 2010, December 2010). You end up with 120 values of average temperature (12 months times 10 years). Maybe change "daily mean" for "monthly average"?
(2) You compute the median of the distribution of January (then February, March and so on) as the value representing the most typical conditions of this particular month. From your 120 monthly averages you first look at the 10 January averages and chose the closest one to the median. You do this for every month which gives you something of the sort: January 2003, February 2010, March 2006, etc.
(3) You extract the hourly values corresponding of the months you've selected and concatenate them into a single new year of hourly data.

This passage has been rephrased: *"Here are the steps to find the most typical (resp. extreme) month for each para :*
   *1 Converting hourly file in daily file: From all the hourly data from all the same calendar months available within the selected period, the daily mean of para is computed;*
   *2 Selecting the typical (extreme) month: For each calendar month, the percentile 50 (resp. 95) of para is calculated over the studied period to find the month which is the closest to the 50 percentile (resp. 95) of para;*
   *3 Extracting hourly data of this typical (extreme) month: Finally, the hourly weather values of this typical (resp. extreme) month are stored in the file of the typical (resp. extreme) year."*

Two questions following these steps:

How do you handle possible jumps in temperature between two months?

The data is smoothed 6h before and 6h after the beginning of each month.

I suppose you compute you percentiles on all the hourly data, so you must take a lot of null values of radiation into account (during night-time). Do you know if this can have an effect on your selection, (especially for the percentile 95) compared to a selection with percentiles computed on day-time values?

No, we computed percentile on daily data. That's why there is a first step to convert hourly data in daily data in order to select the medium or extreme month.

Lines 160-173. I would suggest putting these lines in a table if possible. Maybe something like:
Variable name | Height above ground level | Units

Good idea, thanks. We added a Table 2 at the end of the manuscript.

2.4 Definition of a heatwave and generating the HWE files

Line 178. Do you know how the 18.2 °C and 29.6 °C thresholds were determined? I did not find the information in Tsachoua & Reynders, 2016.

No, it is a good question but we didn't find the answer.

Line 184. Since you screen for heatwaves on every pixels the points I mentioned in section 2.1 regarding the possible representation of cities in MAR (either with an Urban Canopy Model or by replacing them by rocks) is of particular importance since heat waves definition are usually made on rural conditions and do not take into account the urban heat island.

The reviewer is right. We have answered this question above.

Line 187. Do you compute your statistics on the 1980-2014 period (35 years)? Because Ouzeau et al., 2016 use the 1976-2005 period (30 years).

Yes, we think its better to have a longer period to calculate these thresholds but anyway, using 30 or 35 doesn't impact significantly the final value.

Lines 193-194. I think Ouzeau et al., 2016 already had implemented a minimum duration of 5 days for their heatwave detection.

This is ambiguous in Ouzeau et al., so we prefered to clarify it in our manuscript.

Lines 202-204. As Nicolas Heijmans noted, building energy models not only need the values during an extreme event but also the ones that lead to it. Maybe add a sentence explaining how the HWE files can be combined with the TMY and XMY files (depending on what has been decided in the ULiège OCCuPANt project).

More precision here opens the door to additional problems. So we added the sentences below by preferring to remain not precise as each user might combine the files in different ways according to their interests :
*Finally, the HWE files contain only the period corresponding to a heat wave. However, a heat wave can also be dependent on the period preceding and/or following it (depending on the use of the files and the purpose of the users). Thus, a suggestion for users could be that the HWE files can eventually be combined with files of typical or extreme years in order to obtain simulations of a normal year with one or more heat waves. We have not created these files in order to leave it up to the users to combine or not these files and to combine them according to their own constraints and interests.*

4. Conclusion

Line 242. Change "is forced by 4 ESMs" to "is forced by a reanalysis and 3 ESMs".

Indeed, Thanks! This has been corrected in the revised version.

5.1 Appendix A: Example for Liege city

Figures A-1 and A-2. You could add axis labels and variable units.

All figures has been improve to correct this. Thanks.

Technical and editorial corrections

Line 32. If I understood correctly I suggest changing "will warm faster and larger than equatorial regions" to "will warm faster and to higher levels than equatorial regions".

Line 47. Change "This comfort of life" to "This living comfort".
Line 61. Remove "Finally".
Line 70. I would change the order of references so that they are chronologically ascending
Line 73. I would change "6-hourly assimilated" to "assimilated every 6 hours"
Line 79. As Nicolas Heijmans suggested, I would uniformise the SSP notations (SSPx-x.x)
Line 82. Remove "MAR" from "MAR vertical level".
Line 87. Neighbouring with a "u".
Line 125. Remove "y" in "in your case".
Lines 137. Change "the extreme climates" to "the extreme climate".
Table 1. I would remove "annual" from the "Mean annual temperature […]" cell since the average is computed for the whole year and the summer.

We would like to express our sincere thanks to the reviewer for his/her thorough, meticulous, and professional proofreading of our manuscript.

References

Boé, J., Somot, S., Corre, L., and Nabat, P. (2020). Large discrepancies in summer climate change over Europe as projected by global and regional climate models: causes and consequences, Climate Dynamics, 54, 2981–3002
Boé, J. (2021). The physiological effect of CO2 on the hydrological cycle in summer over Europe and land-atmosphere interactions, Climatic Change, 167, 21, https://doi.org/10.1007/s10584-021-03173-2
Gutiérrez, C., Somot, S., Nabat, P., Mallet, M., Corre, L., Meijgaard, E. v., Perpiñán, O., and Gaertner, M. A. (2020). Future evolution of surface solar radiation and photovoltaic potential in Europe: investigating the role of aerosols, Environmental Research Letters, 15, 034 035, https://doi.org/10.1088/1748-9326/ab6666
Ribes, A., Boé, J., Qasmi, S., Dubuisson, B., Douville, H., and Terray, L. (2022). An updated assessment of past and future warming over France based on a regional observational constraint, Earth Syst. Dynam. Discuss. [preprint], https://doi.org/10.5194/esd-2022-7, in review
Schwingshackl, C., Davin, E. L., Hirschi, M., Sørland, S. L., Wartenburger, R., and Seneviratne, S. I. (2019). Regional climate model projections underestimate future warming due to missing plant physiological CO 2 response, Environmental Research Letters, 14, 114 019, https://doi.org/10.1088/1748-9326/ab4949
Wyard, C., Scholzen, C., Fettweis, X., Van Campenhout, J. and François, L. (2017), Decrease in climatic conditions favouring floods in the south-east of Belgium over 1959–2010 using the regional climate model MAR. Int. J. Climatol., 37: 2782-2796. https://doi.org/10.1002/joc.4879
Zelinka, M. D., Myers, T. A., McCoy, D. T., Po-Chedley, S., Caldwell, P. M., Ceppi, P., et al. (2020). Causes of higher climate sensitivity in CMIP6 models. Geophysical Research Letters, 47, e2019GL085782. https://doi.org/10.1029/2019GL085782

**RC2: 'Comment on essd-2021-401', Anonymous Referee #2, 04 Apr 2022**

Thank you for your manuscript about future temperature changes in relation to building simulations. I am not a potential user of this data set, therefore my review may lack in terms of this. However, I found it interesting, although that I do have some comments.

Generally

You use many abbreviations. Please explain all of these when meet first. Example is MAR, but also CMIP6 which you first introduce in line 92 even though it is used much before. There are many more abbreviations than these, so please careful check this to assist the reader.

This has been clarified in the revised version of our manuscript. Thanks.

Be consistent with font type in text and figures

It's not possible because we use a software (NOAA Ferret) to produce figure which have its own specific font types.

Please discuss the sources of errors at a higher level than presented in the paper. This is modelled data, therefore you need to argue why this is useful.
Would other methods than MAR give another results, and would it be better to provide the entire data set used as input, together with the output from the MAR downscaling model?

The idea of this paper is not to have the best estimation of the "true" but homogeneous data both spatially and temporally and to propose a database produced with the MAR model for future climate evolutions in Belgium. Of course, like any climate model, MAR contains biases compared to observations, in particular for precipitation which is the climate field the hardest to simulate by all the models. If the reviewer wishes to know the performance of the MAR model with observations we refer him/her to the papers cited in Section 2.1 where the MAR model is presented. Obviously, there are a variety of other regional models that data users can use. Here, we just propose a new database produced with the MAR model which has been improved these last years to simulate the Belgian climate the closest to observations.

Title

Please do not use abbreviations in title unless you are certain that the reader knows what it means.

The reviewer is right. It has been corrected in the manuscript.

Abstract

Line 20. Which resolution did you have?
This is specified in the manuscript (5km).
What do you mean by 'spatially and temporally homogeneous'?
Compared to weather observation which are punctual data, simulations are spatially (over Belgium) and temporally (over the full simulate period) homogeneous.

Line 24. Normally you don't cite a paper in an abstract.
Please provide information of what the dataset consists of. It is not clear.
This has been corrected.

Introduction

Please explain more clearly what these data is used for. Why is the future data important? Are we designing cooling systems that last 100 years?

The introduction has been adapted to: *"The energy efficiency and living comfort are precisely what motivates the ULiège OCCuPANt project (Impacts Of Climate Change on the indoor environmental and energy PerformAnce of buildiNgs in Belgium during summer, https://www.occupant.uliege.be/). The OCCuPANt project aims to evaluate the energy performance and vulnerability of building inhabitants in the context of climate change. Acquiring reliable current and future climate data is vital in any study related to climate change and defines its quality (Pérez-Andreu and al., 2018). The purpose of this dataset is to propose meteorological data coming from a fine resolution regional climate model over Belgium and the neighbouring regions. These data will then be used to anticipate future climate changes which will influence both the production of heating and cooling demands as well as the electrical grids in larger scales. These changes will require innovations in building design and systems which will necessarily take time. Thus, the more they are anticipated, the better we will find solutions. The use of a regional model allows building spatially and temporally continuous and homogeneous past and future meteorological data according to different warming scenarios for some Belgian cities. This regional model is fed by the ERA-5 reanalysis model (Hersbach et al., 2020) to simulate the past climate (1980-2020) and also by 3 different Earth System Models (ESMs) from the Sixth Coupled Model Intercomparison Project (CMIP6) database (Wu et al., 2019; Tatebe et al., 2019; Gutjahr et al., 2019) to obtain different future projections and associated uncertainties for the same scenario SSP5-8.5.*
*The future climate data are very useful to predict the variations in heating and cooling demands in buildings. The characterization of minimum outdoor temperature under future scenarios is necessary to estimate the heating and cooling season needs. This can result in rethinking the building designs and make them resilient against the impact of climate change. For instance, in heating-dominated region of Belgium, the concept of building design is more on heat retention to decrease the heating energy use during the winter. However, by warming weather conditions in the last decades, such a design concept caused significant overheating problems during the summer. Therefore, it is necessary to predict the future performance of buildings and adapt them to the variations in the outdoor weather conditions. Designing cooling systems that can last for 100 years is challenging. However, it is possible to increase the preparedness of buildings to climate change through passive design strategies, the use of active heating/cooling systems, or both. Both active and passive solutions may need regular replacements over the building life span."*

Methodology

Line 65. Please describe what is meant by reanalysis

This is now explained in the following sentence: *Different kinds of observations (in situ weather observation, radar data, satellites, etc) are 6-hourly assimilated into ERA5 to be the closest to the observed climate. In this way, the simulations of MAR-ERA5 can be considered as the closest simulation to the current observed climate.*
In other words, this is an "earth system model" which assimilate weather observations compared to the earth system models used to project future which do not assimilate any weather observation.

Line 66. I cannot see that 'MAR' is 3-D in Figure 1. Please describe what you mean.

Because Figure 1 only shows the topography which is a 2D field. All other weather variables have data for each vertical level which leads to write that MAR is a 3D model.

Line 78. You don't call this scenario for historical afterwards. Be consistently.

We do not understand the reviewer's remark since this scenario is used in chapter 2.2.3 "evaluation of the MAR simulations". After the evaluation, this historical scenario is effectively no longer used as we make a focus on the future scenarios.

Line 84. 'The spatial resolution of MAR is 5 km…' Isn't the output from the MAR that is 5 km.

The MAR's resolution is 5km so its outputs are also 5km spatial resolution.

Figure 2. Please draw a thick line around Belgium. Not all are familiar with the border of the country.
We do not understand this remark given that the borders of Belgium are well represented in black lines and main rivers in blue lines.

Why are country sides not interesting? Don't they apply this method outside cities?
Because the project in which this research and production of data takes place does not require rural data. But it is quite possible to do it by following the same method as we have outputs everywhere on the integration domain shown in Figure 1
Translated with www.DeepL.com/Translator (free version)

Line 105. You need to explain 'Shared Socioeconomic Pathways' further.

The details of these scenarios are complicated to explain without involving many notions outside of this article. This is why we invite the reviewer to consult the reference paper mentioned in the manuscript. Some  details have been however added to clarify this paragraph:
*"These SSPs are used to develop greenhouse gas emission (GHG) scenarios associated with different climate policies and are used to force each future ESMs. There are 3 main scenarios, namely SSP2-4.5 (intermediate GHG), SSP3-7.0 (high GHG), and SSP5-8.5 (very high GHG), with their respectively increasingly warming for 2100."*

Line 109. What makes the calculations so expensive?

Regional models require a lot of computation time depending on their resolution, their grid, their time frames, their internal physics etc... In this case, the 120-year simulation takes 3 months of simulation without bugs, network problems etc.

Maybe a drawing of the different combinations would help the reader.
3. It is hard to see that red (CSM2-MR) should be higher than the blue color (ESM1.2) referring to line 100-105.

This figure has been improved.

Axis titles and legend is missing.

This has been corrected.

Table 1. Why do you give new names now? Could this be introduced earlier, and applied throughout the paper?

We do not understand what this remark refers to. The name are explained in the text (see Section 2.2.3)

Line 150. Why is it called 'para'
because this is a "parameter" so an easy way to call it is with "para", other ways could have been "P", or "pa" or something like that, but we chose "para" to not confuse with precipitation (P) or pascal (Pa, unit of pressure). "Para" is independent of other weather elements.

and when reading the steps 153-174 it is not clear if you do the steps for each parameter in 'para' or if you combine it.

This is for each para separately as we have specified it in the text: "*Here are the steps to find the most typical (resp. extreme) month for each weather para(meters):*"

Line 188. Please explain Sint, Sdep and Spic, why these names.

We simply used the same names of these thresholds that are used in the reference paper (Ouzeau et al. 2016).

Data set

Very nice with the appendix guiding the user through a data set.
What is TT and SWD?

This is now specified in the manuscript: "*Choosing the parameter that determines the typical or extreme year (TT for temperature and SWD for incident solar radiation), i.e. if you want a year with extreme incident solar radiation (i.e. sunshine), you should select SWDbased.*"

December 31 is missing in e.g. this file Mons-City_TMY1991-2005_hist_MAR-MPI_TTbased.csv. Please check the data that other dates are not missing.

Thank you for pointing out this bug. We have also detected other small errors in the database that we have been a bit revised, but this only concerns a few isolated sets. We will provide a corrected version of this database as soon as the paper will be accepted.

**CC1: 'Comment on essd-2021-401', Nicolas Heijmans, 17 Feb 2022**

General comments:
The paper is very clear, thanks.
And thanks for the data!

We thank Nicolas Heijmans for his review.

Technical comments:
(This is the only strong comment I have :)) I don't understand why you make so strong reference to ISO 15927-4, especially in the abstract (line 22) and in lines 58 and 248, whereas the followed method is not the one of ISO 15927-4. I found this misleading. In the abstract and in line 58, it is said "are generated following the method proposed by the standard ISO15927-4, allowing the reconstruction of typical and extreme years" whereas ISO 15927-4 litterally specify that it can not be applied to generate extreme years ("The procedures in this part of ISO 15927 are not suitable for constructing extreme or semi-extreme years for simulation (…)")
I believe that the reference to ISO 15297-4 should be removed from the abstract, and that the paper should or remove the reference too, or explain why you used a method "based on ISO 15297-4" and not the method of ISO 15297-4 itself. In fact, when I read the procedure (line 154), I even don't see the link with ISO 15297-4!

Nicolas Heijmans is right, the method has been slightly modified. We have therefore corrected all paragraphs discussing the method proposed by the ISO 15927 and this has been corrected to "largely inspired by".

Once again, thanks for the data. However, HWE are not so useful for building simulation, as models have to be "preheated". I mean that the initial conditions of the building can't be guessed, they must be calculated by running the simulation a few weeks before the first day of interest. So, for a heat wave starting e.g. on 01/07, we should have data since 01/06. Another option would be to make available all years, and a table to find the definition of each type of heat waves (but that would make a lot of data!). It would also allow to calculate TMY according to ISO 15297-4 too :)

The reviewer #1 also raised this question, however answering it seems complicated since each research team wishing to work with this data will use different methods and different pre-heat periods as well. So we have added a paragraph to clarify this:
*Finally, the HWE files contain only the period corresponding to a heat wave. However, a heat wave can also be dependent on the period preceding and/or following it (depending on the use of the files and the purpose of the users). Thus, a suggestion for users could be to combine HWE files with files of typical or extreme years in order to obtain simulations of a normal year with one or more heat waves. We have not built these files in order to leave it up to the users to combine or not these files and to prepare them according to their own constraints and interests.*

Possible clarifications:
Line 110 "Thus, since the ESMs do not simulate general atmospheric circulation changes" > that might be clear for climatologist, but it's less clear to usual people like me.

The sentence has been improved: "[…] *since the ESMs do not simulate general atmospheric circulation changes (Eyring, V. et al., 2021) leading that only temperature drives the evolution of the climate. Hence, only the SSP5-8.5 scenario has been calculated to save computational time.*"

Line 234: The reason of having different approaches for leap year could be explained.
Purely editorial comments:

SSP5-8.5 is sometimes written SSP585. Consider to uniformise the notation of SSP's (is the first notation not the right one?).

This has been standardized.

It's not very important, but line 70 says "MAR is initially forced every 6 hours at its lateral boundaries (temperature, wind, and specific humidity)" and line 82 says "The atmospheric variables used to force MAR every 6 hours at each MAR vertical level are temperature, surface pressure, wind, and specific humidity, as well as the sea surface temperature over the North Sea from both ERA5 reanalysis..." > Why not combine those two lines.

We have distinguished the two ones because on the one hand MAR is forced by the reanalyses and on the other hand by the ESMs. Forcing can be sometimes different. In this way, there is no ambiguity.

Line 127: the abreviation "a.g.l." is evident for climatologist, but there is room to write above ground level so that everyone can understand, at least the first time.

This abbreviation has been defined the first time it appears.

Line 164: "Diffuse solar radiation" > Maybe this is the normal denomination for climatologist, but for building simulation, we must be sure to know what is behind. It's quite obvious that it is "solar", but is it on a horizontal surface ? "Diffuse horizontal radiation"

This has been specified in the new Table 2.

(Due to a technical problem, I don't know if those comments are not submitted twice.)

**CC2: 'Comment on essd-2021-401', Nicolas Heijmans, 17 Feb 2022**

By the way, publishing all years would be usefull for other application. For instance, for water management, we don't need a typical year but a series of 20 years.

It's absolutely possible in a further version of the data set but for now this is not the purpose of this data set presented in this article and built in the framework of the OCCUPANT project.

**CC3: 'Minor comment on essd-2021-401', Nicolas Heijmans, 17 Feb 2022**

A last very small point. The following reference could be improved:

European Standard: ISO 15927-4 Hygrothermal performance of buildings — calculation and presentation of climatic data — Part 4: Hourly data for assessing the annual energy use for heating and cooling, 2005.

If you want to refer to an "European" standard (ISO is not European but International), you should refer to:

European Standard: EN ISO 15927-4:2005 Hygrothermal performance of buildings — calculation and presentation of climatic data — Part 4: Hourly data for assessing the annual energy use for heating and cooling (ISO 15927-4:2005)

Thanks a lot. This has been corrected.

---

## Author Response (AR2)

**Topical Editor review**

Authors did not provide adequate response to all reviewer comments. I as editor should not need to intervene at this level. Please make or otherwise respond to each and every correction. Please ensure one full careful reading, with Copernicus guidelines in hand, of final manuscript.

Line 35: Western Europe is not a "temperate oceanic region".

*According to Köppen-Geiger-Peel climate classification (Peel et al., 2007), a large part of Western Europe is in a "Cfb" climate (i.e. Group "C" for temperate climate, Group "f" for without dry season and Group "b" for warm summer) which can be summarised as oceanic temperate climate, knowing that without the influence of the ocean some seasons could be drier. This term "oceanic" is indeed open to discussion. Furthermore, we can also discuss the definition of "Western Europe" which does not have a clear universal definition, for example, sometimes Spain is included in Western Europe sometimes not, and the eastern boundary of Western Europe is blurred and variable depending on the definition. Thus, to avoid any ambiguity, we have deleted the term "oceanic", which allows us to really encompass as many countries as possible while remaining in line with the Köppen-Geiger-Peel climate classification.*

Lines 131-132: This sentence, about ESMs, fails on both accuracy and clarity.

*This sentence has been rephrased in two sentences:*
*"However, as the ESMs are not able to represent general circulation changes (Eyring, V. et al., 2021), the use of one scenario or another will not cause more blocking anticyclones leading to more persistent heatwaves for example. Thus, whichever scenario is used will only reflect temperature changes in relation to its GHG concentrations."*

Lines 136 to 147: This entire paragraph seems confusing at best, random and inaccurate at worst. Please start fresh on a clear rational paragraph. Define 'computational' needs or limits and "these issues".

*This paragraph has been rephrased in:*
*"Hence, only the SSP5-8.5 scenario has been calculated and the other scenarios (SSP3-7.0 and SSP2-4.5) are derived from the MAR simulations forced by the ESMs using the SSP5-8.5 scenario since the warming rates from lower scenarios are included in the scenario SSP5-8.5 but for a different earlier time period. Thus, for each scenario (SSP3-7.0 and SSP2-4.5), the equivalent warming period in the SSP5-8.5 scenario has been found according to these 3 steps:*
*1) The raw 2m annual mean temperature of each ESM and each scenario has been aggregated over Belgium to the horizon 2100;*
*2) For each ESM, the equivalent 20 years period from the SSP5-8.5 scenario has been chosen as the period with the closest mean and the closest interannual variability of the 2m annual temperature compared to the future 20 years period (i.e. 2021-2040, 2041-2060, …) from the two other scenarios (SSP3-7.5 and SSP2-4.5);*
*3) Once the equivalent warming period has been identified, the data of this period is extracted out of the SSP5-8.5 forced MAR simulations for both SSP3-7.5 and SSP2-4.5 scenarios. For example, the data of MAR-MPI for SSP3-7.0 over2081-2100 are the outputs of MAR-MPI using SSP5-8.5 over the period 2066-2085.*
*This method is open to discussion for several reasons, firstly because the climate does not react in a linear way to an increase of GHG flowing through the different SSP scenarios. Moreover, with an equal warming rate but different periods, the Earth, Atmosphere and Ocean systems will not have the same (spatial and temporal) responses due to their inertia. Despite these precautions, this methodology allows us on the one hand to derive a quick estimation without additional computer*

*time. On the other hand, it remains valid as a first approximation, especially since the most interesting weather variable in this study is temperature, which is, by construction, the least sensitive to these issues."*

Line 150: For a manuscript published in 2022, 1980 to 2014 does not represent "present". It instead represents (as already defined) historical. I did not find authors' answers to reviewer complaints on this dichotomy (present vs historical) clear or compelling. Also noted by at least one reviewer. Please resolve in a careful thoughtful useful manner.

We have changed this sentence in: "[…] *it is necessary to evaluate them over the overlapping period between the ERA5 reanalyses and the historical scenario (namely 1980-2014 )."*
And we have added in section 2.2.2 this sentence to avoid any ambiguity: "*Finally, it should be noted that the historical scenario mentioned in Section 2.1 is forced by the greenhouse gas concentrations observed over the period 1980-2014."*

Line 153: a few lines earlier (line 147) a reader learned that only temperature mattered. Here we learn that 2m temp plus surface radiation represent the most important factors. Very confusing?

We have adapted this sentence to: "*As the most important variable for this database is the temperature at 2m above ground level (a.g.l.) and an important secondary variable is the incoming solar radiation, the mean and standard deviation of these data over the period 1980-2014 and over the Belgian territory are compared in Tab. 1 and Fig. 3."*

Line 178: 'parameter' abbreviated as "para". Odd, bogus, confusing, ineffective. Please remove and restore.

This term has been replaced in the whole manuscript without abbreviation.

Line 197: having visited many cities and regions in Belgium, this reader has yet to encounter what most of us would consider 'highlands'. If authors want to use (mis-use?) this term, they at least need to define the elevations. Figure 2 does not help.

We have corrected in: "[…] *in the highlands (with altitudes above 300m in Figure 2) [...]"*

Geographic standards exist for city names in any/every country. Please assure that use of names here meets those standards, by cross-listing if necessary.

We have corrected some city names according to the city names employed by the National Geographic Institute of Belgium (https://topomapviewer.ngi.be/).

Line 225: dependent on the period preceding and/or following it (according to use of the files … Please correct as suggested or otherwise change wording to make intention clear.

We have corrected this sentence in: "*Finally, the HWE files contain only the period corresponding to a heatwave event. However, depending on the purpose of the users, the effects of a heatwave can also be dependent on the period preceding and/or following it."*

However we do not know which suggestion you are referring to as this whole paragraph was added after the 1st round of review.

Figure 2: Topography units in C? Please correct.

This is a surprising remark because we do not see any topography with a unit of degrees Celsius. To make sure we are using the same figure, we have corrected this Figure 2 according to the remark above (about city names) and we took the opportunity to add an "m" next to the colour scale to avoid any ambiguity.

Figure 3: unacceptable in present weak format.

Fig. 2, Fig. 3, Fig. A1 and Fig. A2 have been revised and the image format of Fig. 3, Fig. A1 and Fig. A2 have been changed in ".tiff" instead ".png". These new figures have been added at the end of the main document but also in a PDF supplement file as suggested.

Somewhere authors need to describe in full text the ESM sources: e.g Beijing Climate Centre, Max Planck Institute, etc. At least one reviewer already made this request.

We have added a Table A1 to summarize all these information.

Many typographical, punctuation and abbreviation errors, plus errors of meaning and context. Overall, not well written. Type setters will fix some of these errors but authors need to first conduct full careful review and edit following closely to Copernicus publication guidelines.

The whole text was checked for punctuation, grammar and other errors according to the Copernicus guidelines. We have noted that the Copernicus guidelines prohibit abbreviation in titles. We have removed these abbreviations except for one: the regional model MAR.
We absolutely want to keep this abbreviation in the title like all our articles (see Agosta et al, 2019; Doutreloup et al. 2019, 2021; Fettweis et al., 2013; Kittel, 2021 and many others) in order to make this model known. However, to be as explicit as possible we have changed the title to :
*"Historical and Future Weather Data for Dynamic Building Simulations in Belgium using the regional climate model MAR: Typical & Extreme Meteorological Year and Heatwaves"*.
We hope that this wish and this title modification can be accepted.

**Notification to the authors:**

1. Appendix figures should have standard headers (not "A-2", but "A2")
We have corrected these references it in the figure captions but also in the manuscript.

2. For the next revision, please pay attention to the quality of the supplement images. In addition, I would recommend that you combine all images and their captions into a single PDF supplement file.

Fig. 2, Fig. 3, Fig. A1 and Fig. A2 have been revised and the image format of Fig. 3, Fig. A1 and Fig. A2 have been changed in ".tiff" instead ".png". These new figures have been added at the end of the main document but also in a PDF supplement file as suggested.

**References :**
Peel, M. C., Finlayson, B. L., and McMahon, T. A.: Updated world map of the Köppen-Geiger climate classification, Hydrol. Earth Syst. Sci., 11, 1633–1644, https://doi.org/10.5194/hess-11-1633-2007, 2007.

---

## Author Response (AR3)

05 Jun 2022
Topical Editor decision: Publish subject to technical corrections
by David Carlson

Comments to the author:
Thank you for positive changes. I comment on a few remaining issues below; any changes can occur at proof stage.

Fig 2: In presentation, the rightmost numbers of the topographic legend appear clipped, so that the final 0 appears instead as a C. E.g. middle value of 400 m appears instead as 40C. Small correction in image registration should easily fix this display problem.

We have changed the figure by increasing border in order to not clipped values.

Fig 3: Black lines (MAR-ERA5) appear for the period 1980 to 2020 but, in contrast to the legend text, they do not appear bold? Here the period of ERA5 forcing appears as 1980 to 2020 (2019?), not as 1980 to 2014?

We have changed the captions according to your remark by correcting the text in parenthesis: « *(in black lines between 1980 and 2020)* »

Fig 4 mentions (at end of legend) a web version of this article. Not mentioned elsewhere. If such a version exists (e.g. separate from the ESSD eventually-published version), authors should include its URL in data availability section (Section 3)?

We have changed the captions according to your remark by correcting the end of the caption by : « […] the reader is referred to Ouzeau et al.(2016).) »